# Abscopal Effect on Bone Metastases from Solid Tumors: A Systematic Review and Retrospective Analysis of Challenge within a Challenge

**DOI:** 10.3390/biomedicines11041157

**Published:** 2023-04-12

**Authors:** Miriam Tomaciello, Miriam Conte, Francesca Romana Montinaro, Arianna Sabatini, Giorgia Cunicella, Federico Di Giammarco, Paolo Tini, Giovanni Luca Gravina, Enrico Cortesi, Giuseppe Minniti, Giuseppe De Vincentis, Viviana Frantellizzi, Francesco Marampon

**Affiliations:** 1Department of Radiological Sciences, Oncology and Anatomical Pathology, Division of Radiotherapy, Sapienza University of Rome, 00161 Rome, Italy; 2Department of Radiological Sciences, Oncology and Anatomical Pathology, Division of Nuclear Medicine, Sapienza University of Rome, 00161 Rome, Italy; 3Department of Radiological Sciences, Oncology and Anatomical Pathology, Division of Oncology, Sapienza University of Rome, 00161 Rome, Italy; 4Radiation Oncology Unit, Department of Medicine, Surgery and Neurosciences, University of Siena, 53100 Siena, Italy; 5Department of Biotechnological and Applied Clinical Sciences, University of L’Aquila, 67100 L’Aquila, Italy; 6IRCCS Neuromed, 86077 Pozzilli, Italy

**Keywords:** abscopal effect, radiotherapy, bone scintigraphy, bone metastases

## Abstract

Background: Abscopal effect (AE) describes the ability of radiotherapy (RT) to induce immune-mediated responses in nonirradiated distant metastasis. Bone represents the third most frequent site of metastasis and an immunologically favorable environment for the proliferation of cancer cells. We revised the literature, searching documented cases of AE involving bone metastases (BMs) and evaluated the incidence of AE involving BMs in patients requiring palliative RT on BMs or non-BMs treated at our department. Methods: Articles published in the PubMed/MEDLINE database were selected using the following search criteria: ((abscopal effect)) AND ((metastases)). Patients with BMs, who underwent performed bone scintigraphy before and at least 2–3 months after RT, were selected and screened between January 2015 and July 2022. AE was defined as an objective response according to the scan bone index for at least one nonirradiated metastasis at a distance > 10 cm from the irradiated lesion. The primary endpoint was the rate of AE on BMs. Results: Ten cases experiencing AE of BMs were identified from the literature and eight among our patients. Conclusions: The analysis performed here suggests the use of hypofractionated radiotherapy as the only triggering factor for AE of BMs through the activation of the immune response.

## 1. Introduction

Radiation therapy (RT) uses ionizing radiation (IR) to directly kill cancer cells [1]. However, RT has been shown to also kill nonirradiated metastatic cancer cells distant from the irradiated site, namely, the abscopal effect (AE), a reflection of the ability of IR to elicit an anticancer immune response [2]. The mechanisms triggering AE are still not largely understood. Briefly, irradiated tumor cells release danger signals, expose tumor-associated antigens, and attract/activate antigen-presenting cells, finally leading to the priming of cytotoxic CD8-expressing T cells (CTLCD8+). CTLCD8+, the most powerful effectors in the anticancer immune response, can then migrate from lymph nodes to distant tumor sites, finally mediating AE [2,3]. Notably, despite the increasing use of hypo-fractionated or stereotactic ablative RT, known to improve IR’s ability to promote immune responses to tumors [4], AE has remained a rare phenomenon for a long time, with only 46 cases reported up to 2014 [5]. However, the introduction of immune checkpoint inhibitors (ICIs), shown to boost the RT-induced immune response [6,7] and to be potentiated by RT [8], has drastically increased the number of cases reporting AE [9,10,11,12,13,14,15,16,17,18,19,20,21,22,23,24,25,26,27,28,29,30,31,32,33,34,35,36,37,38,39,40,41,42,43,44,45,46,47,48,49,50,51,52,53,54,55,56,57,58,59,60,61,62,63,64,65,66], definitively turning it from myth to reality [67].

Bone is the third most common metastasis site, with lung, kidney, breast, prostate cancer, and melanoma accounting for approximately 80% of all bone metastases (BMs) [68]. RT is the standard of care only for pain relief or lytic lesions or in oligometastatic cancer, not in all patients [69], while ICIs results are largely inefficient [70,71,72,73]. Thus, the control of BMs seems to be related more to the directly cytotoxic action of the IR rather than to the immune-mediated one, suggesting that eliciting the AE of BMs is more complex than in visceral metastases. Notably, to date, there is no literature review describing how frequent AE of BMs is and what conditions favor its implementation.

Here, we have systematically reviewed the literature for AE cases involving BMs, and we conducted a retrospective study involving a large sample of patients treated at our radiotherapy center. The aim was to clarify whether BMs can effectively be considered immunologically cold, and what could be the best strategies capable of boosting the immune response induced by RT towards BMs.

## 2. Materials and Methods

### 2.1. Search Strategy and Studies Selection

A systematic electronic search was performed in the PubMed/MEDLINE database for articles published between 1 January 1973, and 31 December 2022, using the following search query: ((abscopal effect)) AND ((metastases)). Studies were included if published in English, reporting clinical results of patients treated with RT on metastatic sites, in combination or not with other treatments, describing cases of AE involving BMs, and distant > 10 cm from the radiation treatment site [74]. An independent review of the abstract and full-text of the articles was performed by 2 reviewers (M. T., M. C., and F. R. M.). The articles that fulfilled the inclusion criteria were shortlisted for a final systematic review.

### 2.2. Data Extraction and Methodological Quality Assessment of the Included Studies

General data, such as the authors, journal, year of publication, country, and study design, were collected for each included paper. The methodological quality assessment was realized using the Quality Assessment of Diagnostic accuracy Studies-2 (QUADAS2) tool. The data extraction and quality evaluation were conducted by two reviewers, and the possible disagreements were resolved by discussion among the researchers.

### 2.3. Retrospective Analysis of Our Patient Population

A retrospective data extraction was performed between January 2015 and July 2022 at the Policlinico Umberto I Hospital, Sapienza University of Rome. Patients with BMs from solid tumors were considered. The inclusion criteria were (1) age > 18 years; (2) radiologically proven BMs; (3) RT for at least one metastatic site; (4) the presence of at least one nonirradiated metastasis which was >10 cm away from the irradiated target volume [75]; (5) multiple sites must have been treated simultaneously; and (6) the presence of bone scintigraphy before and at least 2–3 months after RT. A clinical examination was performed and blood samples were collected before, during, and after RT. The RT-targeted irradiation volume was defined as the planning target volume (PTV) covered by at least 95% of the prescribed dose. The following patient characteristics were recorded at the beginning of the RT: RT total dose, RT dose per fraction, RT duration, localization of target volumes, number of irradiated sites treated simultaneously, and types and modalities of the last systemic treatment. The AE was defined as an objective response according to the scan bone index (BSI) for at least one nonirradiated metastasis at a distance > 10 cm from the irradiated lesion. The BSI was defined as the overall percentage of BMs identified on scintigraphic images relative to the total bone mass [75,76]. The values of the BSI were assessed through DICOM Analyzer Scintigraphy Software (DASciS software), an automatic tool for bone scan quantitation developed in Java by an engineering team from the Sapienza University of Rome. It is helpful in assessing the disease burden in patients with secondary bone involvement and describes how the BSI evolves in response to various treatments. This instrument is compatible with all of the most popular operating systems (i.g., Windows, MacOS, and Linux) and has an effective graphical user interface (GUI). The DICOM format of Gamma Cameras images was analyzed using DASciS software. The operator selects the disease portion, and on the basis of the intensity of the pixel, the software automatically selects all pixels in the image with intensities equal to or greater than the one chosen. All pixels are grouped into multiple regions of interest (ROIs) according to the approach of Suzuki [77] and Green’s theorem [78]. The physician modified the contrast intensity of the images and excluded some of them that were characterized by an increased fixation not associated with malignancy, such as recent fractures and bladder activity due to the fact of the tracer’s clearance. The DASciS outputs included the total percentage of ill regions computed compared to the remaining whole-body bone mass. The OpenCV library is used for image processing [79], and the DICOM files were processed using the open-source PixelMed library. The program saves the processed statistical data in a CSV file or other extensions.

## 3. Results

### 3.1. Systematic Review Analysis

The literature search resulted in 273 articles. Based on a screening process, 10 were selected for revision. Within the 263 excluded articles; 8 were reviews; 1 was an editorial article; 39 were preclinical studies; 82 dealt with an abscopal effect, not on BMs; in 6 articles the AE was a consequence of chemotherapy–immunotherapy treatments without the involvement of radiotherapy; and the remaining 117 did not respond to our endpoints. The systematic search process is summarized in Figure 1. The methodological quality of the included studies was excellent quality. All selected works satisfied each of the QUADAS-2 domains, and all studies obtained a low concern of bias.

Ten articles, summarized in Table 1, reported AE occurring on BMs [11,21,23,27,31,52,64,75,76,77].

The median reported age was 66.5 years (age range: 30–94 years). Four patients suffered lung cancer, the other five patients resulted affected, respectively, by endometrial, head and neck, breast, kidney, or liver cancer, while the primary tumor was unknown in one patient. Nine patients received external beam radiation therapy (EBRT), seven on non-BMs and two on BMs. Zero patients received conventional schedule (CRT), five patients a hypofractionated schedule (HFRT), and two stereotactic irradiation (SBRT), while no information was available for one patient. The median reported radiation dose was 66 Gy (range: 26–225 Gy), with a median dose per fraction of 8.5 Gy per fraction (range: 2–26 Gy). EBRT was exclusive in six patients, concomitant to immunotherapy in two patients, and conventional chemotherapy in one patient. One patient received brachytherapy (BRT) on BMs, preceded and followed by immunotherapy. The median documented time to notice the AE was 3.8 months (range: 1 week–12 months), and the median reported follow-up was 30 months (range: 12–72 months).

### 3.2. Methodological Quality of the Retrieved Studies

#### Our Retrospective Analysis

Between January 2015 and July 2022, 743 patients were treated with palliative RT at our institution. A total of eight patients met the inclusion criteria and were included in the analysis (Figure 1). The patient demographics and clinical characteristics are detailed in Table 2.

Four patients were women, and four were men; the median age was 66.12 ± 15 years (range: 37–84 years). Four patients were diagnosed with prostate cancer (PCa), three with breast cancer (BCa), and one with small cell neuroendocrine carcinoma (SNCE). All patients received EBRT. One patient received whole brain irradiation, receiving a total dose of 25 Gy in 10 fractions (2.5 Gy/fraction). Seven patients received palliative RT on BMs: three patients on one BM, one patient on two BMs, two patients on four BMs, and one patient on five BMs. Seventy-seven percent of irradiated BMs were vertebral while twenty-three percent other bones. All patients received a total dose of 20 Gy in five fractions. No patient had undergone chemotherapy or ICI treatments in the six months before, during, or immediately after RT. All patients with prostate cancer BM treated with RT on BMs were on androgen deprivation therapy (ADT), started on average 22 months earlier (range: 18–25 years), and then continued after radiation.

Eight patients (100%) experienced a total of 21 AE on BMs: two patients on one (4.7%) BM, two patients on two (9.4%) BMS, two patients on three (14.3%) BMs, one patient on four (14.3%) BMs, and one patient on five (71.6%) BMs. Concerning the localization, AE occurred in one (4.7%) vertebra, two (9.4%) skulls, four (18.8%) ribs, two (9.4%) femurs, two (9.4%) sternums, three (14.1%) both humeri, and three (14.1%) tibias. The baseline BSI of the BMs who underwent AE was 0.109 ± 0.079 before RT and 0.042 ± 0.041 after RT, with a mean difference of 0.067 ± 0.051, equal to 61.5% less. Figure 2 shows some of the BMs subjected to AE, before and after RT (Figure 3).

## 4. Discussion

Combining radiotherapy (RT) and immune checkpoint inhibitors (ICIs) has definitively turned the abscopal effect (AE) from myth to reality [67], where there have been 46 cases of AE described in 45 years up to 2014 [5] and another 58 cases reported in the last 8 years [9,10,11,12,13,14,15,16,17,18,19,20,21,22,23,24,25,26,27,28,29,30,31,32,33,34,35,36,37,38,39,40,41,42,43,44,45,46,47,48,49,50,51,52,53,54,55,56,57,58,59,60,61,62,63,64,65,66]. However, only 10 case reports describe AE of bone metastases (BMs) [11,21,23,27,31,52,64,75,76,77]. Thus, considering that bones are the third most by frequency among metastatic sites [68], triggering an AE of BMs seems to be a challenge within a challenge, and identifying the best strategies to elicit AE in BMs is strategic as it would benefit many patients.

Based on the data reported in the literature, hypofractionated RT, particularly 30 Gy in 10 fractions, seems to facilitate AE of BMs according to what has already been suggested for other metastatic sites [7,72]. Only three patients received ICI which, therefore, seems to favor but not be indispensable to the induction of AE in BMs. However, the optimal sequencing seems to be ICIs concomitant or pre- and post-RT ICIs, confirming that ICIs should be not administered only before RT [8]. Interestingly, the largest of the patients experiencing AE of BM received RT on visceral lesions. This suggests that the type of irradiated lesion may facilitate the appearance of BMs AE and that BMs irradiation would not produce any immunosensitization of other metastases. Surprisingly, analyzing the eight cases of the AE of BMs identified in our department, we had partially conflicting evidence. In accordance with the literature, the patients experiencing AE of BMs received hypofractionated RT. In particular, the fractionation used in almost all cases was five fractions of 4 Gy each. However, no patient had undergone ICI therapy either before, during, or after RT, and seven out of eight patients had undergone RT on BMs. Notably, all patients with BMs from prostate cancer were treated with androgen deprivation therapy (ADT) during RT. ADT has been shown to promote strong adaptive antitumor T- and B-cell responses [79], while we do not know whether it has the same effect in metastases.

Bone represents an ideal environment for cancer cells to escape from immunosurveillance [80,81,82]. Cancer cells, before homing to the bone, release immunosuppressive cytokines, recruit, and activate myeloid-derived suppressor cells, immunosuppressive regulatory T cells, and M2 macrophages, mainly involved in anti-inflammatory responses [83]. Furthermore, after homing, cancer cells suppress the expression of MHC class I molecules, resulting in the inability to present the antigen [7,83], type I IFN signaling, an immune escape mechanism associated with an increased risk of BMs [7], and interact with several types of bone-resident cells, finally sustaining the evasion from the anticancer immunosurveillance [8,84]. These immunosuppressive characteristics of BMs have been recently shown to attenuate the efficacy of ICIs [70] and could be potentially responsible for the low rate of AE after irradiation of BMs.

Bone scintigraphy (BSI) represents the first choice between the imaging techniques to detect bone metastases. Since CT is a valid tool to identify bone destruction that could be evident after a relevant cortical destruction, the sensitivity of this method is slow in early malignant bone involvement [81]. Moreover, malignant marrow infiltration is not easily detected by CT, despite the fact the tumor-infiltrated marrow is more attenuated than normal marrow [82]. Bone scan, indeed, remains a sensitive, widespread, and affordable imaging modality with the lower cost [85,86].

The introduction of BSI allowed for the evaluation of bone disease’s load yet used for the assessment of skeletal disease’s burden at starting of radium 223 and for the estimation of overall survival in a previous study conducted by our research group [87].

In this study, BSI has represented an ulterior instrument for bone recurrences that permitted to identify the AEs of BMs.

Despite the remarkable results, some limits affected this study, such as the lack of a multicentric evaluation and the smaller population. However, this work is a preliminary appraisal and supplementary studies are in progress.

## 5. Conclusions

The abscopal effect after the irradiation of BMs represents a real phenomenon, already documented and further studied in our experience. The evidence currently present in the literature does not suggest any specific strategy to increase the occurrence probability of AEs in BMs. In this context, bone scan is constituted as a valid instrument to detect the AEs of BMs. The low cost and availability of this imaging modality could permit to obtain astonishing results in oncological research.

## Figures and Tables

**Figure 1 biomedicines-11-01157-f001:**
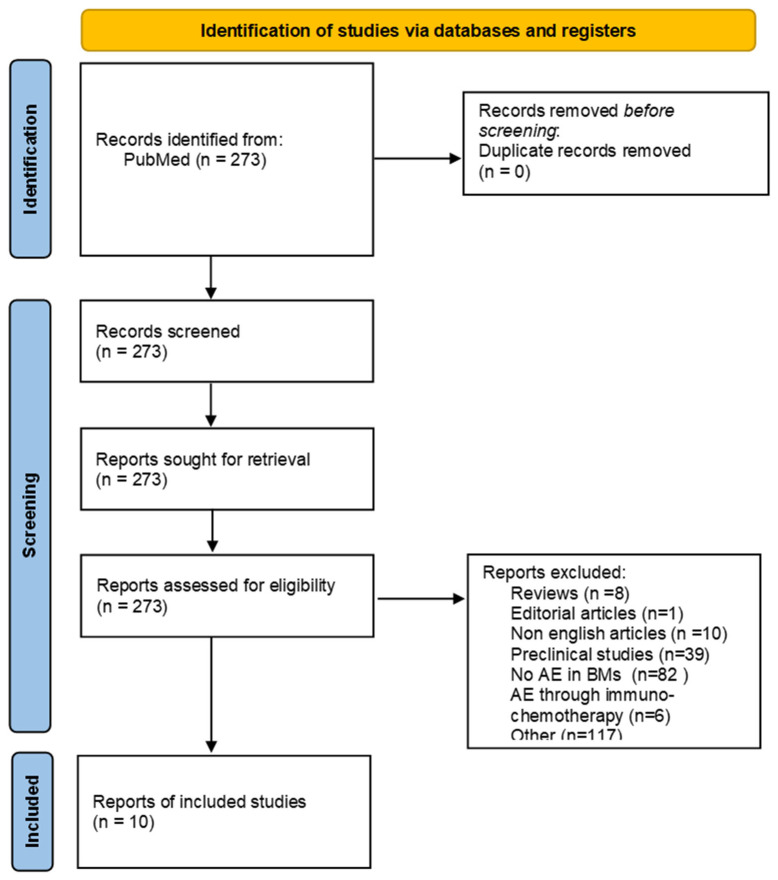
Representation of the PRISMA workflow for the selection of articles.

**Figure 2 biomedicines-11-01157-f002:**
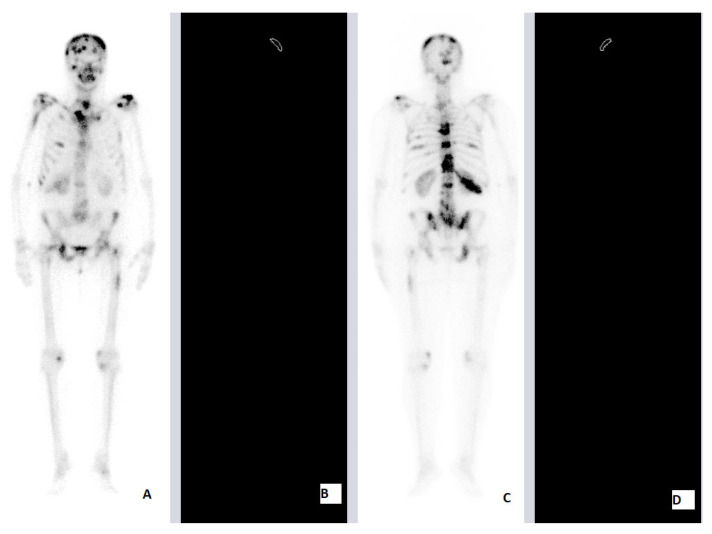
Female breast cancer patients with RT sites that were thoracic and lumbar vertebrae. Note the metastasis on the skull case. (**A**) Bone scintigraphy, anterior; (**B**) Dascis software, anterior, ROI on skull; (**C**) bone scintigraphy, posterior; (**D**) Dascis software, posterior, ROI on skull.

**Figure 3 biomedicines-11-01157-f003:**
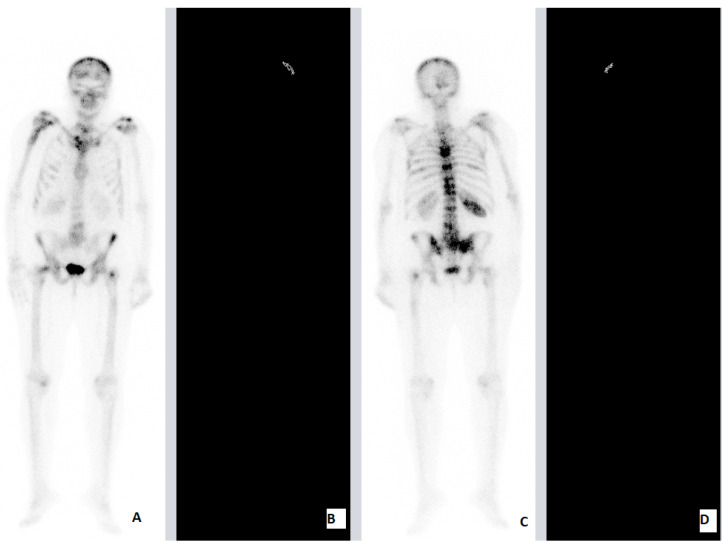
Same female breast cancer patients as in Figure 2 after RT. The RT sites were thoracic and lumbar vertebrae. Note the reduction of metastasis on the skull. (**A**) Bone scintigraphy, anterior; (**B**) Dascis software, anterior, ROI on skull; (**C**) bone scintigraphy, posterior; (**D**) Dascis software, posterior, ROI on skull.

**Table 1 biomedicines-11-01157-t001:** Characteristic of the patients and treatments of case reports from the literature describing the AE of BMs.

Study	Primary Tumor	Age	Gender	RT Type	Total Dose (Dose/Fraction)	Site of Metastasis Different from Bone	Site of RT	Systemic Therapy (Close to RT)		BMs Site of AE	Time for AE (Months)	Follow-Up Duration (Months)
								Type	Timing			
Yano M. et al. (2020) [11]	ECs	65	F	HFRT	30 Gy (10 Gy)	LG, M-LNMs	Pelvis	ICI	Before Concomitant	Scr	7	14
Mazzaschi G. et al. (2021) [19]	H&N	66	M	SIB	70 Gy (2 Gy)	N/A	Oph Hph Ln	CHT	Before	Lhs Hmr	2	72
					40 Gy (2 Gy)	N/A	LNMs					
Vilinovszki O. et al. (2021) [23]	NSCLC	81	F	HFRT	36 Gy (12 Gy)	Cervical LNMs	LG	--	--	12th TV, 4th LV, LG, LNMs	0.25	25
Ishikawa Y. et al. (2022) [27]	Unknown	57	M	HFRT	39 Gy (13 Gy)	Cervical LNMs	9th TV	--	--	8th Lhs Rib, Rhs Ilm, LGs, LNMs	1	30
Sakaguchi T. et al. (2022) [31]	SCC	94	M	HFRT	30 Gy (10 Gy)	M-LNMs	IC	--	--	Axial vertebrae	0.25	NA
Siva S. et al. (2013) [52]	NSCLC	78	F	SBRT	26 Gy (26 Gy)	AG	LG	--	--	Rhs Hmr, AG	12	15
Leung H. W. et al. (2018) [62]	BCa	65	F	SBRT	225 Gy (15 Gy)	A-LNMs	Breast	--	--	8th TV	3	48
Nam S. W. et al. (2005) [75]	HCC	64	M	HFRT	30 Gy (6 Gy)	N/A	Skull			Skull, Stn, Ribs, Lvr	3	24
Golden E. B. et al. (2013) [76]	ADK	64	M	Lvr	30 Gy (NA)	Cervical and M-LNMs	lvr	ICI	Concomitant	Lhs scr, LG, Lvr, Hilar LNMs	2.5	12
Suzuki G. et al. (2019) [77]	Renal cell carcinoma	30	F	BRT	7 Gy + 9.2 Gy + 8.5 Gy + 7.9 Gy	LG, Rhs ovary, Hilar and M-LNMs, lvr	IC	ICI	Before After	TV	3	NA

A-LNMs: axillar lymph nodes; ADT: androgen deprivation therapy; AG: adrenal gland; BCa: breast cancer; BRT: brachytherapy; CRT: conventional RT; CHT: chemotherapy; Clc: clavicula; ECs: endometrial carcinosarcoma; F: female; Fmr: femurs; HCC: hepatocellular carcinoma; H&N: head and neck; HFRT: hypofractionated RT; Hph: hypopharynx; Hmr: humerus; IC: iliac crest; ICIs: immune checkpoint inhibitors; Ilm: ilium; LG: lung; lhs: left; Ln: larynx; LV: lumbar vertebrae; Lvr: liver; LNMs: lymph nodes; M: male; M-LNMs: mediastinal lymph nodes; N/A: not available; NSCLC: non-small cell lung cancer; Oph: oropharynx; rhs: right; SCC: squamous cell lung cancer; Scr: sacrum; SBRT: stereotactic RT; Stn: sternum; TV: thoracic vertebrae; SIB: simultaneous integrated boost; SJ: sacroiliac joint; Tb: tibia.

**Table 2 biomedicines-11-01157-t002:** Characteristic of the patients and treatments experiencing AE of BMs after RT in our department.

Primary Tumor	Age	Gender	RT Type	Total Dose (Dose/Fraction)	Site of Metastasis	Site of RT	Systemic Therapy (Close to RT)		BMs Site of AE	Time for AE (Months)	Follow-Up Duration (Months)
							Type	Timing			
SNCE	69	F	HFRT	25 Gy (2.5 Gy)	Brain 12th TV 1st, 3rd, 5th LV 3rd, 7th Rhs Rib Rhs Tb Lhs Hmr LNMs	Brain			Lhs Hmr 7th Rhs Rib	4	6
PCa	84	M	HFRT	20 Gy (4 Gy)	7th, 8th, 9th, 10th TV Stn Lhs Tb Rhs Tb	7th, 8th, 9th, 10th TV	ADT	Concomitant	Stn Lhs Tb Rhs Tb	2	5
PCa	72	M	HFRT	20 Gy (4 Gy)	11th TV Lhs Fmr Rhs Fmr Lhs Hmr Rhs Hmr	11th TV Lhs Fmr Rhs Fmr	ADT	Concomitant	Lhs Hmr Rhs Hmr	2	6
BCa	60	F	HFRT	20 Gy (4 Gy)	Brain LG 5th, 10th, 11th, 12th TV 1st, 2nd LV Rhs Ischium Pelvis Lhs Fmr Rhs Fmr 12th Rhs Rib Rhs Clc LNMs	10th, 11th, 12th TV 1st, 2nd LV			SkC 12th Rhs Rib Rhs Clc Lhs Fmr	2	8
PCa	83	M	HFRT	20 Gy (4 Gy)	Lhs SJ Stn Lhs Fmr	Lhs SJ	ADT	Concomitant	Stn Lhs Fmr	1.5	4
PCa	75	M	HFRT	20 Gy (4 Gy)	5th, 6th, 7th, 8th TV 2nd Rhs Rib Rhs SJ	5th, 6th, 7th, 8th TV	ADT	Concomitant	2nd Rhs Rib Rhs Fmr	1.5	4
BCa	58	F	HFRT	20 Gy (4 Gy)	Kidney Brain AG 10th, 11th TV Pelvis Lhs Fmr Rhs Fmr SKC	Rhs Fms			SKC	3	5
BCa	37	F	HFRT	20 Gy (4 Gy)	3rd, 4th, 5th CV 4th, 5th LV Stn 9th, 12th Rhs Rib Scr Pelvis LNMs	Sacrum			4th LV	3.5	6

PCa: prostate cancer; BCa: breast cancer; SNCE: neuroendocrine; M: male; F: female; HFRT: hypofractionated; lhs: left; rhs: right; TV: thoracic vertebrae; LV: lumbar vertebrae; Scr: sacrum; Tb: tibia; Hmr: humerus; Stn: sternum; LG: lung; Fmr: femurs; SJ: sacroiliac joint; Clc: clavicula; AG: adrenal gland; LNMs: lymph nodes; SKC: skull case; ADT: androgen deprivation therapy.

## Data Availability

No new data were generated.

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
