# Peer review of "Abscopal Effect on Bone Metastases from Solid Tumors: A Systematic Review and Retrospective Analysis of Challenge within a Challenge"

_biomedicines, 2023, doi:10.3390/biomedicines11041157_

Round 1
Reviewer 1 Report
Thank you for the authors for this interesting paper. However, some aspects must be clarified.
Line 75: Radiation therapy is standard of care only for pain relief or lytic lesions or in oligometastatic cancer, not in all patients. Please clarify this in the text.
Table 1: Please add the years of selected publifications. The timing of systemic therapy could be written after the therapy, so space for other columns could be spared. RT type should be classified more accurately, such as hypofractionated, stereotactic, conventional, etc. EBRT and BRT should be explained under the table.
Is the abscopal effect a term that has been known "always"? To a larger use it has been taken only in recent years. Should the literature search include also other terms? Has this caused bias?
Lines 171-173 seem to be internal discussion and should be removed.
Table 2: The used abbreviations should be explained. As these are all fine, is this table necessary?
Table 3 is unnecessary as the content can be said easily by few words (line 198).
Table 4: The table should be formulated similar to table 1. Now RT type and follow-up duration are missing, while site of metastasis is not reported in table 1. Site of RT is in different location than in table 1.
Table 4: Please tell how long the Decapeptyl was in use in those patients with this treatment.
Figures 1-3 were not included in the submission and thus cannot be reviewed.
How many patients were excluded because of missing bone scan after radiotherapy? Does this cause biases?
Line 255: Hypofractionated therapy does not mean the common palliative radiotherapy 5 x 4 Gy. Please correct/clarify. This is repeated in the line 291.
Line 275: The sensitivity might be low, but not slow. Please correct.
Line 288: There seems to be confusion on terms. The abscopal effect after irradiation of metastases on BMs should be correct wording.
Language should checked thoroughout the paper.
Author Response
We want to thank you for your review, valuable comments and suggestions.
- Line 75: Radiation therapy is standard of care only for pain relief or lytic lesions or in oligometastatic cancer, not in all patients. Please clarify this in the text.
The sentence suggested has been included in the text (Now Line 75).
- Table 1: Please add the years of selected publications. The timing of systemic therapy could be written after the therapy, so space for other columns could be spared. RT type should be classified more accurately, such as hypofractionated, stereotactic, conventional, etc. EBRT and BRT should be explained under the table.
The years of selected publications have been added. The timing of systemic therapy has been written after the therapy. RT type has been classified more accurately.
- Is the abscopal effect a term that has been known "always"? To a larger use it has been taken only in recent years. Should the literature search include also other terms? Has this caused bias?
The term "abscopal effect" is the one "always" used, since the first description of the event (doi: 10.1259/0007-1285-26-305-234.). For this reason, the bibliographic research has not included other terms, without causing any bias.
- Lines 171-173 seem to be internal discussion and should be removed.
Lines 171-173 have been removed
- Table 2: The used abbreviations should be explained. As these are all fine, is this table necessary?
Abbreviations have been included. In accordance with the reviewer's suggestion, the table has been removed and made available as "data not shown 1".
- Table 3 is unnecessary as the content can be said easily by few words (line 198).
In accordance with the reviewer's suggestion, the table has been removed and made available as "data not shown 2".
- Table 4: The table should be formulated similar to table 1. Now RT type and follow-up duration are missing, while site of metastasis is not reported in table 1. Site of RT is in different location than in table 1.
Table 4 has been formulated like table 1. RT type and follow-up duration are now included in table 4. Site of metastasis have been now reported in table 1 (Se nei manoscritti non sono scritti gli altri siti di metastasi mettiamo N/A…tabella 1 dove ho messo i ??). Site of RT in table 4 is now in the same location than in table 1.
- Table 4: Please tell how long the Decapeptyl was in use in those patients with this treatment.
The information required has been included in the paragraph “Our retrospective analysis” (All patients with prostate cancer BM treated with RT on BMs were on androgen deprivation therapy (ADT), started on average 22 months earlier (range, 18–25 years), and then continued after radiation)
- Figures 1-3 were not included in the submission and thus cannot be reviewed.
We apologize to the reviewer for this error. The figures have now been inserted correctly.
- How many patients were excluded because of missing bone scan after radiotherapy? Does this cause biases?
Most of the patients were excluded precisely because of the lack of a bone scintigraphy. Performing bone scintigraphy in the follow-up of a metastatic patient is not exactly a common practice, nor is its use to verify any AEs on BM. The reduced number of patients included is an expression of this and we do not believe it could have generated any bias. On the contrary, the evidence presented here on the ability of bone scintigraphy to clearly show the efficacy of radiotherapy treatment in the induction of Ae on BMs should prompt the use of this diagnostic investigation during the follow-up of these patients.
- Line 255: Hypofractionated therapy does not mean the common palliative radiotherapy 5 x 4 Gy. Please correct/clarify. This is repeated in the line 291.
The sentence in line 255 has been now fixed, clarifying the concept that the fractionation used in almost all cases was 5 fractions of 4 Gy each (Now Line 259). We did not find any other repetition. The line 291 reports:” The bone scintigraphy (BSI) represents the first choice between the imaging techniques to detect bone metastases [92,93]. Since CT is a valid tool to identify bone destruction which could be evident after a relevant cortical destruction, the sensitivity of this method is slow in early malignant bone involvement [94]”.
- Line 275: The sensitivity might be low, but not slow. Please correct.
The term “slow” has now been corrected with “low” (Now Line 279).
- Line 288: There seems to be confusion on terms. The abscopal effect after irradiation of metastases on BMs should be correct wording.
The sentence has been fixed.
- Language should checked thoroughout the paper.
Language has been checked.
Reviewer 2 Report
I do not agree entirely with the statement “hypofractionate RT seems to be the only certainty” , there are not sufficient data in literature and in your casuistry to fully agree with this conclusion partly because some pts were treated with ICIs too.
*Fig 1, 2 , 3 are not present in the manuscript
Additional minor flaws as follows:
* rows 171-173: ??? it looks as a reminder for the authors
* Table 1 and 4 are difficult to read. Use for instance abbreviations for vertebras, left/right and so on
* Table 2 : It is preferable to explain P(atients) I(ndex test) R(eference standard) FT(flow and timing) in the table or in the legend.
It is self explanatory that the green checkmark means “low risk” but the risk level must be specified.
*Table 4:
pt #4 brain metastases and AE in the skull? brain or bone?
pt#5 lift /left
*row 231: homers
*Ref.16: First name instead of last name
Author Response
We want to thank you for your review, valuable comments and suggestions.
- I do not agree entirely with the statement “hypofractionate RT seems to be the only certainty” , there are not sufficient data in literature and in your casuistry to fully agree with this conclusion partly because some pts were treated with ICIs too.
The sentence “The use of hypofractionated RT seems to be the only certainty.” has been removed.
- Fig 1, 2 , 3 are not present in the manuscript.
We apologize to the reviewer for this error. The figures have now been inserted correctly.
- rows 171-173: ??? it looks as a reminder for the authors
Lines 171-173 have been removed.
- Table 1 and 4 are difficult to read. Use for instance abbreviations for vertebras, left/right and so on.
Abbreviations have been now included and table 1 and 4 formated also following the suggestions of the reviewer 1.
- Table 2 : It is preferable to explain P(atients) I(ndex test) R(eference standard) FT(flow and timing) in the table or in the legend.
Following reviewer 1's suggestion, table 2 has been removed and now presented as "Data not shown". P(patients) I(index test) R(reference standard) FT(flow and timing) have been legend.
- It is self explanatory that the green checkmark means “low risk” but the risk level must be specified.
Following the suggestions of reviewer 1, the table has been removed and will show as data not shown. However, the risk level must have ben now specified.
- Table 4: pt #4 brain metastases and AE in the skull? brain or bone?
We meant the skull case. The term Skull Case has been now included.
- Table 4: mpt#5 lift /left
Left has been fixed.
- row 231: homers
Homer has been fixed.
- Ref.16: First name instead of last name
The reference number 16 has been fixed.
Reviewer 3 Report
This is an interesting manuscript on a relevant topic. The study and the review are well conducted and comprehensive.
I wish the Authors clarify the two different situations: radiotherapy alone and radiotherapy combined with medical treatment/immunotherapy either pre or post or concomitant. Why to consider altogether and not to divide in different groups? Medical treatment could be considered a confounding factor if not well described and justified
Author Response
We want to thank you for your review, valuable comments and suggestions.
This is an interesting manuscript on a relevant topic. The study and the review are well conducted and comprehensive. I wish the Authors clarify the two different situations: radiotherapy alone and radiotherapy combined with medical treatment/immunotherapy either pre or post or concomitant. Why to consider altogether and not to divide in different groups? Medical treatment could be considered a confounding factor if not well described and justified
We want to thank the reviewer for pointing out such a critical point. Both in the case of the literature review and in the description of our patients, the tables and the text referring to them describe whether radiotherapy was concomitant with other treatments or exclusive. To clarify the data reported, the tables have been updated with the breakdown of patients, as suggested by the reviewer.
Round 2
Reviewer 2 Report
I appreciated the changes in text and the addition of three figures. The paper can be published in such form.